# Assessment of Antioxidant and Anticancer Activities of Microgreen Alga *Chlorella vulgaris* and Its Blend with Different Vitamins

**DOI:** 10.3390/molecules27051602

**Published:** 2022-02-28

**Authors:** Ragaa A. Hamouda, Amera Abd El Latif, Ebtihal M. Elkaw, Amenah S. Alotaibi, Asma Massad Alenzi, Hanafy A. Hamza

**Affiliations:** 1Department of Biology, College of Sciences and Arts, Khulis, University of Jeddah, Jeddah 21959, Saudi Arabia; 2Department of Microbial Biotechnology, Genetic Engineering and Biotechnology Research Institute, University of Sadat City, Sadat 32897, Egypt; ebtihal.alkaw@gmail.com (E.M.E.); hanafi.hamza@gebri.usc.edu.eg (H.A.H.); 3Department of Pharmacology, Faculty of Veterinary Medicine, Kafrelsheikh University, Kafrelsheikh 33516, Egypt; amirashehata12@yahoo.com; 4Biology Department, College of Sciences, Tabuk University, Tabuk 71491, Saudi Arabia; a_alotaibi@ut.edu.sa (A.S.A.); amalanazi@ut.edu.sa (A.M.A.)

**Keywords:** vitamins, *Chlorella vulgaris*, carbohydrate, antioxidant, antitumor

## Abstract

There is a very vital antioxidant extracted from microgreen alga. *Chlorella vulgaris* has major advantages and requires high yield worldwide. Some microalgae require vitamins for their growth promotion. This study was held to determine the impact of different vitamins including Thiamine (B1), Riboflavin (B2), Pyridoxine (B6), and Ascorbic acid (c) at concentrations of 0.02, 0.04, 0.06, and 0.08 mg/L of each. Each vitamin was added to the BG11 growth medium to determine the effect on growth, total carbohydrate, total protein, pigments content, antioxidant activities of *Chlorella vulgaris*. Moreover, antitumor effects of methanol extract of *C. vulgaris* without and with the supplement of thiamine against Human prostate cancer (PC-3), Hepatocellular carcinoma (HEPG-2), Colorectal carcinoma (HCT-116) and Epitheliod Carcinoma (Hela) was estimated in vitro. *C. vulgaris* supplemented with various vitamins showed a significant increase in biomass, pigment content, total protein, and total carbohydrates in comparison to the control. Thiamine was the best vitamin influencing as an antioxidant. *C. vulgaris* supplemented with thiamine had high antitumor effects in vitro. So, it’s necessary to add vitamins to BG11 media for enhancement of the growth and metabolites.

## 1. Introduction

Cancer lingers to be one of the principal sources of death in the Earth. Researchers sustained searching for harmless and further actual chemoprevention and care is required for the enhancement of the proficiency and to lower the cost for cancer treatments [1]. Microalgae have usually become part of the daily diet, specifically in East Asian countries; algae are an attractive research topic because of their active contents used to treat cancer [2]. Microalgae are usually used as an additive in food to enrich the nutrient of food and/or mend the health of beings due to phytochemical contents and bioactive molecules. The protein composition of microalgae is the chief goal for the untraditional source of protein [3]. Moreover, the microalgae which could synthesize vital amino acids to humans and animals compares beneficially with that of other sources of proteins [4]. In recent years, many researchers studied the assembly of extracellular antibiotic metabolites by marine algae, and the researchers discovered that a large number of antifungal agents are present in green microalgae extracts [5]. Ultimately, microalgae are almost an available supply of natural antioxidants due to their huge biodiversity, more varied than plants. They enclose various biologically vigorous composites that are consumed as a source of food, feed, and medicine [6]. Microalgae are offered as a replacement to the molecular pharming system. [7]. Natural antioxidants exposed greater antioxidant productivity than synthetic antioxidants [8] Algae represent a sources of natural antioxidant compounds such as carotenoid, vitamins, phenolic compounds, and phycobilins, that have different applications in various areas such as pharmaceutical, nutraceutical, medicine and food industries [9]. Microalgae are an important source of pharmacologically vigorous metabolites with antineoplastic, antitumor, antibacterial, antifungal, and antiviral activities [6]. *Chlorella* is considered one of the oldest microorganisms on earth due to its sphere form and very stable cell wall. *Chlorella* sp. is a type of unicellular eukaryotic green microalga [10]. Taiwan is one of the major countries to produce Chlorella-related productions. The world annual sales of *Chlorella* sp. are more than US$ 38 billion [11]. *C. vulgaris* which related to the marine microalgae have been found to contain the huge number of highly nutritious effective substances which are valuable to the human body and exert various biological effects such as pigments, proteins, unsaturated fatty acids, squalene, polysaccharides, etc. [12]. It is widely used in popular food additives and it is considered a potential source for healthy food in many countries and is effective in improving overall health and well-being [13,14]. It is rich in proteins, which involve all essential amino acids requisite for human growth and wellbeing [15]. It contains chlorophyll-*a* and *b*, and primary carotenoids, such as lutein and β-carotene. Moreover, it is considered a functional whole food with lower cholesterol due to providing large amounts of dietary fibers, nucleic acids, carbohydrates, vitamins, fatty acids, minerals, and chlorophyll [16]. *Chlorella vulgaris* supplementation has been reported to have certain promising physiological properties, specifically protecting against the immunosuppressive effect of stress and gastric ulcer formation [17]. Furthermore, it showed anti-inflammatory, immune-modulatory hepatoprotective, anti-diabetic, antihypertensive, antioxidative and hypocholesterolemic, and anticancer or antitumor activities [18,19,20,21]. Vitamins are a chemically unrelated organic compound that cannot be synthesized in adequate quantities by beings and therefore must be blend to the diet. Nine vitamins (B complex and non-complex (vitamin C) are classified as water-soluble, whereas four vitamins (vitamins A, D, K, and E) are classified as fat-soluble. They perform various roles such as cofactor of various enzymes, the regulator of mineral metabolism and antioxidant activities, etc.; also concerned in the general metabolism of the organism. Many research illustrated algal species that demand diverse combinations of three B vitamins: vitamin B1 (thiamine), vitamin B12 (cobalamin), and vitaminB7 (biotin). Cruz-López and Maske [22] reported that the maximum growth rate of *Lingulodinium polyedrum* (dinoflagellate) was obtained when acquiring B1 and B12 vitamins from the associated bacterial community. Algae were incubated with vitamins showed a growth rate increase with higher vitamins concentrations, meanwhile algal cell volume decrease [23]. There are very few studies related to the effects of various vitamins supplementations on algal growth and productivity. This study was aimed to compare the effects of different vitamins added to the BG11 growth medium on the *Chlorella vulgaris* growth, Chlorophyll-*a*, Chlorophyll *b*, carotenoid, protein, carbohydrates contents, and antioxidant activity. In addition, the antitumor effect of *C. vulgaris* and *C. vulgaris* blend with thiamine at different concentrations was determined.

## 2. Results

The results showed the effect of different vitamins supplementations (Thiamine (B1), Riboflavin (B2), Pyridoxine (B6), Ascorbic acid (C)), used at four different concentrations (0.02, 0.04, 0.06 and 0.08 µg/ L) on the biomass production (Optical density), total photosynthetic pigments contents, antioxidant (DPPH) activity, total protein and carbohydrate contents of *C. vulgaris* during 18 days of cultivation.

### 2.1. The Effect of Vitamins Supplementations on C. vulgaris Biomass Production

Results obtained in Figure 1a show the optical density of *C. vulgaris* supplemented with different concentrations of Pyridoxal (vitamin B6) at 660 nm. The growth of *C. vulgaris* was increased within three days of inoculation with all tested concentrations of Pyridoxal (vitamin B6). The best concentrations were 0.08 µg/L of Pyridoxal followed by 0.06 µg/L. Meanwhile the control *C. vulgaris* (without vitamin supplementation) produced the least amount of biomass compared with the other concentrations of Pyridoxal. Figure 1b demonstrated the effect of different concentrations of Ascorbic acid (vitamin C) on *C. vulgaris* growth within 18 days of incubation. Moderate growth was obtained at the first three days of incubation. The maximum growth was obtained with the highest concentrations (0.08 µg/L) until the 14 days of incubations, but after 14 days the elevations of growth were obtained by the following Ascorbic acid (vitamin C) concentrations 0.02, 0.04, 0.08, and 0.06 µg/L and control, respectively. The elevation of *C. vulgaris* growth was observed in the first three days of incubation with all the tested concentrations of Thiamine. From three to seven days the stationary phase of *C. vulgaris* occurred with 0.06 µg/L of Thiamine. Moderate elevations of *C. vulgaris* growth occurred with the following concentrations, 0.02, 0.04, and 0.08 µg/L of Thiamine at 3 to 7 days of incubations, meanwhile, the elevations of growth were obtained after seven days of *C. vulgaris* incubation until 16th day of incubation with 0.08, 0.08 µg/L and control, respectively, Figure 1c. BG 11 media when supplemented with riboflavin (vitamin B2), the growth of *C. vulgaris* was increased, Figure 1d. High elevations of *C. vulgaris* growth were obtained in the first three days of cultivations with all concentrations of riboflavin. The 0.02 µg/L was the best concentration of riboflavin that caused high elevation of growth at 10 to 20 days followed by 0.08, 0.04 and 0.06 µg/L, respectively, in compared with control.

### 2.2. The Effect of Vitamins Supplementations on the Photosynthetic Pigments of C. vulgaris

The present results indicated that there were significant differences in chlorophyll-a, chlorophyll-b and carotenoid pigments at the different sampling days except for chlorophyll-b in the control group showed non-significant differences. Moreover, significant differences were observed in the pigments contents for different concentrations of Pyridoxal at different experimental days being at range from 0.19 (0.08 µg/L concentrate after 3 days) to 3.06 (control group after 11 days) for chlorophyll-a and from 0.84 (0.08 µg/L concentrate after 3 days) to 6.42 (0.02 µg/L concentrate after 16 days) for chlorophyll-b, while they varied from 5.20 (0.08 µg/L concentrate after 3 days) to 24.12 (0.02 µg/L concentrate after 11 days) for Carotenoid (Table 1).

It appears that the sampling time had no significant influence on chlorophyll-a content at 0.06 and 0.08 concentrations of ascorbic acid; chlorophyll-b for the control group and 0.06 ascorbic acid concentration; Carotenoid for 0.06 and 0.08 µg/L ascorbic acid concentrations. In contrast, a significant effect of sampling time was observed on all pigments at 0.02 and 0.04 µg/L ascorbic acid concentration; chlorophyll-b at 0.08 µg/L concentration; chlorophyll-b and Carotenoid for the control group. Furthermore, significant differences appeared in the pigments’ contents for different concentrations of ascorbic acid at different experimental days varied from 1.10 to 5.56 for chlorophyll-a, from 0.36 to 4.56 for chlorophyll-b, and from 5.36 to 22.91 for Carotenoid as presented in Table 2.

Table 3 demonstrates the effect of different concentrations of thiamine and days of incbation on the pigments content of *C. vulgaris*. As presented in Table 1, no-significant effects of sampling time were observed on chlorophyll-b content for control group, 0.04 and 0.08 µg/L Thiamine concentrations; Carotenoid at 0.02 and 0.06 µg/L concentrations, while significant differences were recorded for chlorophyll-a content at different concentrations; chlorophyll-*b* at 0.02, 0.06 µg/L concentrations; Carotenoid at 0.04 and 0.08 µg/L concentrations. In addition, significant differences were observed in the pigments’ contents for different concentrations of Thiamine at different experimental days except 0.02 and 0.04 µg/L concentrations after 3 and 16 days; concentration 0.06 µg/L after 11 and 16 days. Minimum estimates were 0.03, 0.08, and 3.77 µg/L, while the maximum estimates were 4.05, 4.26, and 29.27 µg/L for chlorophyll-a, chlorophyll-b, and carotenoids, respectively.

Results in Table 4 indicate that there was a significant effect of sampling time from day 3 to day 20 post the beginning of the experiment on chlorophyll-*a*, chlorophyll-b, and Carotenoid pigments except for chlorophyll-b in the control group showed non-significant differences. In addition, non-significant differences were observed in the pigments’ contents for different concentrations of riboflavin at different experimental days except control group at different days, 0.02 µg/L concentration after 3, 11, and 20 days, 0.06 after 3 and 16 days and 0.08 µg/L concentrations after 9 and 16 days. Minimum estimates were 0.04, 0.40, and 0.04 µg/L, while the maximum estimates were 5.30, 4.48, and 17.48 for chlorophyll-*a*, chlorophyll-*b* and carotenoids, respectively.

### 2.3. Effect of Different Concentrations of the Tested Vitamins on Carbohydrate and Proteins Contents of C. vulgaris

Results in Table 5 indicated that there were highly significant differences in carbohydrates and proteins contents under the effect of the different concentrations of the tested vitamins. The best concentration of vitamins for enhancement of carbohydrate content of *C. vulgaris* was 0.08 µg/L for all the tested vitamins. Meanwhile, the best concentration was different among vitamins concerning their effect on proteins content as the best concentration was 0.08 µg/L Thiamine, 0.06 Riboflavin, and Pyridoxal. Meanwhile, Ascorbic acids diminished the protein content of *C. vulgaris*. concerning protein values, minimum values were coupled with 0.04 concentration for all vitamins, while the maximum values were observed at 0.08 concentration for Thiamine, 0.02 for Ascorbic acid; 0.06 for Riboflavin and Pyridoxal.

### 2.4. Antioxidant Activities of C. vulgaris and C. vulgaris Supplemented with Different Concentrations of the Tested Vitamins

DPPH (Antioxidant assay) 1,1-Diphenyl-2-picryl-hydrazyl, showed highly significant differences in the antioxidant activity of *C. vulgaris* under the effect of different concentrations of the tested vitamins being at range from 42.80 to 56.26% Ascorbic acid, 42.26 to 56.13% Riboflavin, 42.13 to73.60% Thiamine, and 12.53 to 23.20% for Pyridoxal, Figure 2. The result shows the *C. vulgaris* blends with Pyridoxal gives lower antioxidant activities with all tested concentrations, and also in compared with the effect of other vitamins and concentrations. Meanwhile, the Thiamine concentrations 0.04 and 0.08 µg/L possessed higher antioxidants activity in compared to control.

### 2.5. Antitumor Effect of C. vulgaris and C. vulgaris Supplemented with Different Concentrations of Thiamine (Vit. B1)

The effects of different concentrations of *C. vulgaris* and *C. vulgaris* blended with 0.08 µg/mL of Thiamine on the inhibition of various cancer cells were summarized in Table 6. Results revealed that methanol extracts of *C. vulgaris* and *C. vulgaris* supplemented with Thiamine at different concentrations (100, 200, 400, 600 and 800 µg/mL) affect inhibition of various cancer cells (HEPG-2, HCT-116, Hela and PC-3). However, *C. vulgaris* supplemented with Thiamine gives better results compared with *C. vulgaris* without Thiamine supplementation. The higher concentrations were most effective against cancer cells

### 2.6. FT-IR

The FT-IR analyzes of *C. vulgaris* biomass (control) and blended with thiamin embodied various absorption peaks Figure 3. The peaks were, with *C. vulgaris* control 3404, 2970, 2925, 2856, 1655, 1549, 1408, 1384, 1054, 711 and 568 cm^−1^ which has moved to 3449, 2959, 2954, 2853, 2768, 1646, 1384, 1076, 875, 831, 600 and 564 cm^−1^. In comparison between results of *C. vulgaris* biomass (control) and blended with thiamin, bands 3404 cm^−1^ in *C. vulgaris* are moved to 3449 cm^−1^, the same site denoted the O-H stretching vibration (3500–3200 cm^−1^), that related to occurred of alcohols, phenols. The bands 2970 and 2925 changed to 2959 and 2923 in *C. vulgaris* (control), and *C. vulgaris* to blend with Thiamine. The peaks range from 3300–2500 cm^−1^ are representing O–H stretching vibration occasion of carboxylic acids. The band 1655 was moved to 1646 and 1045 changed to 1076. The band 2426 was showed only with *C. vulgaris* blended with Thiamine, and the two bands 1549 and 1048 were showed only with *C. vulgaris* (control), these results denoted the change in chemical structure of *C. vulgaris* blended with Thiamine.

## 3. Discussion

Vitamins play vital roles in the development of the contents of microalgae [24]. Algae required various nutrients in addition to carbon dioxide, light, and water for growth. Meanwhile, for faster growth, numerous microalgae require the supply of vitamins such as thiamine, biotin, or cobalamine because they cannot synthesize them [25,26]. Certain types of microalgae grow fast and efficiently in the presence of prokaryotes that produce vitamin B that is absorbed by microalgae [27]. Vitamins perform a vital role in the promotion of the primary productivity of the marine ecosystem and induction of microalgae bloom [28,29]. Vitamin B1 was the first cofactor that affects in microalgae growth and also plays a major role in carbohydrate and protein metabolism [30,31,32]. A large number of microalgae demand an external supply of thiamine [24]. Total chlorophyll-*a* of *Cyclotella nana* increased with all concentrations of vitamin B12 [23]. Moreover, Menzel and Spaeth [33] reported that vitamins have a vital role in marine ecology not only in the capability to impact in yields but also in affecting in growth rate. When *Cyclotella nana* grew in different concentrations of vitamin B12, the cell numbers were little changed during the first 2 days, the best concentration of vitamin B12 was 3 ng [23]. However, Algae were significantly increased when vitamins were found in the growth media [33]. Total chlorophyll-*a* was elevated daily when *Cyclotella nana* grew at various concentrations of vitamin B12 but in *Monochrysis* *lutheri*, chlorophyll-*a,* elevated in the first 3 days of incubations [23]. Furthermore, Lwoff and Dusi [34] investigated that some cryptophyta demands vitamins such as thiamine as a growth factor. Menzel and Spaeth [33] showed that diatom blooms grow moderately when cobalamine was supplemented at the highest concentrations. In addition, several studies correlate algal yields and vitamin concentrations [35,36]. Harmful algal blooms were increased with vitamins concentration in water in a similar manner as amino acids and protein [37]. More than 70% of microalgae need supplementations of one or more vitamins [38]. Microgreen algae *Volvox aureus* and *Pandorina morum* required thiamine and vitamin B12, *Desmidium swartzii* required vitamin B12, *Synura petersenii* required vitamin B12, and biotin for growth [38,39]. Supplementation of *Chu* 10 D medium with vitamins B1, B6, B7 and B12 produced higher biomasses of *Chlorella vulgaris* that were possibly used as feedstock for the productions of biofuels [40]. Vitamin B6 enhances the growth of *Chaetoceros* *calcitrans* [41]. The growth rate of *Haematococcus* was increased when the growth media were supplemented with 50 mg/L vitamin B12 [42]. Desouky [43] reported that the total carbohydrate contents of *C. vulgaris* were significantly elevated when supplemented with 200 ppm of riboflavin (vitamin B2). Vitamins are essential for the continued growth of some algae and also an adverse effect of some heavy metals [43,44,45]. Thiochrome (a natural metabolite of thiamine) can enhance the reproduction of *Chlorella vulgaris* [46]. Total carbohydrate content was significantly elevated when 200 ppm of ascorbic (vitamin C) acid or thiamine (vitamin B1) were added for different levels of either CoCl_2_ or NiCl_2_ to *Chlorella vulgaris* [47]. The *Chlorella vulgaris* cell number, dry weight, total pigments content, and total carbohydrates were significantly increased when inoculated in different levels of Pb^+2^ and supplemented with 200 ppm riboflavin [43]. Total carbohydrates and total proteins increased significantly when *Chlamydomonas reinhardtii* was supplemented with thiamine and different concentrations (25, 50 and 75 mM) of NaCl [48]. The free amino acid contents of *Scenedesmus obliquus* were significantly increased when supplemented with 200 ppm pyridoxine and riboflavin [49]. *C. vulgaris* supplemented with Thiamine is a potential source of natural antioxidants [50]. *Chlorella vulgaris* has antioxidant activities and anti-cholesterol [51]. The addition of Thiamine promoted the oxidative damage of Cu^+2^ on *C. vulgaris* growth and improved growth, pigment contents [52]. The antioxidant enzymes of *Chlorella vulgaris* were increased when exposed to UV-B [53]. The higher antioxidant activities were present in Chlorella methanolic extract [54]. Magnetic fields enhance the growth and antioxidant activities of *Chlorella vulgaris* [55]. *Chlorella vulgaris* has antioxidant and anticancer activities [56]. Hot water extract of *Chlorella vulgaris* decreased the number of viable HepG2 cells due to increased DNA damage and apoptosis [57]. Nanocellulose and the Au/cellulose nanocomposite derived from Chlorella vulgaris posses’ significant cytotoxicity against lung cancer cells (A549) due to improve the relative expression of p53 gene, while reduced that of the Raf-1 gene [58]. In the FT-IR spectroscopy analysis of *C. vulgaris* biomass and supplemented with Thiamine there was the change in the peak position and also there were new peaks and disappear other peaks in the alga supplemented with thiamine, that denoted that modification of algal structure after treatments and hence effect in properties and metabolites [59]. Structural changes particular components were investigated using vibrational spectroscopy [60]. Fourier transform infrared (FTIR) spectra measured the change of macromolecular composition of *Microcystis aeruginosa* and *Protoceratium reticulatum* in response to the treatments [61].

## 4. Materials and Methods

### 4.1. Tested Alga

*Chlorella vulgaris* was attained from our lab (GEBRI), University of Sadat City, Egypt, and used as a test organism. The medium was BG11 (Table 7) nutritive culture for the enrichment and growing of the tested alga. *Chlorella vulgaris* culture subject (control) and four different vitamins concentration were supplemented to BG11 medium, Thiamine (B1), Riboflavin (B2), Pyridoxine (B_6_), Ascorbic acid (c). Vitamin concentrations were adjusted to 0.02, 0.04, 0.06 and 0.08 µg/L. Two. Each vitamin was added to the culture media after the medium sterilization in autoclave and the vitamin sterilize by Millipore filter 45 µ pore, due to the vitamins may be decompose high temperature in autoclave. The cultures were put at a west-facing window receiving natural daylight at a temperature 30 ± 2 °C and shaken moderately thrice a day to prevent clumping. Culture growth was estimated by optical density at 660 nm by using Shanghai Unico UV-2000 spectrophotometer.

### 4.2. Preparation of Algal Extracts

Algal cultures, *C. vulgaris,* and *C. vulgaris* supplemented with different concentrations of the tested vitamins were put in the previously mentioned conditions. In a log phase, algal biomass was collected, washing and dried. Equal to one g of dried alga was mixed with 100 mL of methanol for 4 h in magnetic stirrer, after 4 h the algal mixture was filtrated through Whatman No.1 filter paper, and then dried by vacuum at 50 °C using a rotary evaporator.

### 4.3. Estimation of Pigments Content of C. vulgaris

Five mL of homogenized *C. vulgaris* suspension were centrifuged for 10 min at 4000 rpm. The extractions of pigments were obtained by using 5 mL 90% acetone from the pellets. The tubes were placed in the dark for 24 h. After that sample was centrifuged for 15 min at 5000 rpm and the supernatant was collected. The absorbance was read at 662, 645, and 470 nm (A662) against 90% acetone as blank by using Shanghai Unico UV-2000 spectrophotometer. The calculation of pigments according to the following equations:
Chlorophyll-*a* (µg·mL^−1^) = 11.75·A662 − 2.350·A645
Chlorophyll-*b* (µg·mL^−1^) = 18.61·A645 − 3.960·A662
Carotenoids (µg·mL^−1^) = 1000 A470 − 2.270·Ca − 81.4 Cb/227

### 4.4. Determination of Protein and Carbohydrate Content of C. vulgaris

Protein contents of *C. vulgaris* supplemented with various concentrations of the tested vitamins were determined using Folin phenol reagent according to the method adopted by Lowry et al. [62], carbohydrate contents were determined by the Anthrone method of Hedge and Hofreiter [63].

### 4.5. Evaluation of the Antioxidant Activity of C. vulgaris In Vitro

#### 4.5.1. DPPH Free Radicals Scavenging Assay

The antioxidant activity of *C. vulgaris* extracts and *C, vulgaris* extracts supplemented with the tested vitamins at various concentrations (0.02, 0.04, 0.06, and 0.08 µg/L) were tested by 2, 2,2-diphenyl-1-picrylhydrazyl (DPPH) test. One ml of 0.03 g DPPH in 50 mL methanol was mixed to 1 mL of methanol extract *C. vulgaris* culture subject (control), and along with four different vitamins concentration and incubated for 30 min at room temperature, the absorbance was measured at 517 nm by using Unico UV-2000 spectrophotometer. The antioxidant activity was calculated according to the following equation.
$$\mathrm{Antioxidant}\;\mathrm{activities}\%=\left(\frac{B-S}{B}\right)*100$$
(S) The absorbance of sample, (B) Absorbance of blank

#### 4.5.2. Determination of the Antitumor Activities of *C. vulgaris*

##### Cell Line

Four human tumor cell lines namely; Human prostate cancer (PC-3), Hepatocellular carcinoma (HEPG-2), Colorectal carcinoma (HCT-116), and Epitheliod Carcinoma (Hela) were used. The cell lines were obtained from ATCC via Holding company for biological products and vaccines (VACSERA), Cairo, Egypt.

##### Chemical Reagents

The RPMI-1640 medium, MTT and DMSO (sigma co., St. Louis, MI, USA), Fetal Bovine serum (GIBCO, Cramlington, UK) were used.

##### MTT Assay

The cell lines declared above were used to investigate the suppression effects of methanol extracts of *C. vulgaris* and *C. vulgaris* blended with 0.08 µg/L of thiamine on cell growth using the MTT assay [64,65]. (The methanol extracts of *C. vulgaris* and *C. vulgaris* blended with 0.08 µg/L of thiamine were evaporated by Rotary evaporator, followed by dissolved in Dimethyl sulfoxide (DMSO), to prepare the following concentration (100, 200, 400, 600, and 800 µg/mL). The colorimetric assay is measured at absorbance of 570 nm using a plate reader (EXL 800, New York, NY, USA). The relative cell viability in percentage was calculated as (A570 of treated samples/A570 of untreated sample) × 100.

### 4.6. FT-IR Analysis

Fourier transform infrared (FTIR) spectroscopy was used to estimate the presences and change of the functional groups in *C. vulgaris* and *C. vulgaris* blended with Thiamine [66].

### 4.7. Statistical Analysis

Fixed effects included vitamin concentrate, pigment, and time were investigated according to Proc ANOVA using a statistical analysis system (SAS, 2012), and a *p*-value of ≤0.05 was considered to indicate statistical significance. The differences between means were detected by Duncan‘s Multiple Range Test [67].

## 5. Conclusions

*C. vulgaris* has applications in food, antibiotics, pharmaceutical compounds and is also used in biofuels such as biodiesel and bioethanol. In this work, we have tested the effects of adding various vitamins at different concentrations to *C. vulgaris* growth media on the *C. vulgaris* growth criteria (Optical density, total photosynthetic pigments, antioxidant (DPPH) activity and total protein, carbohydrate contents) during 18 days of cultivation compared with control *C. vulgaris* (without vitamins supplementations). The results confirmed that all the supplemented vitamins have a significant effect on alga growth (increasing *C. vulgaris* biomass) and the best vitamin was thiamine at a concentration of 0.08 µg/L. In addition, *C. vulgaris* without and with thiamine have anticancer properties when tested in vitro however, *C. vulgaris* supplemented with thiamine was more effective.

## Figures and Tables

**Figure 1 molecules-27-01602-f001:**
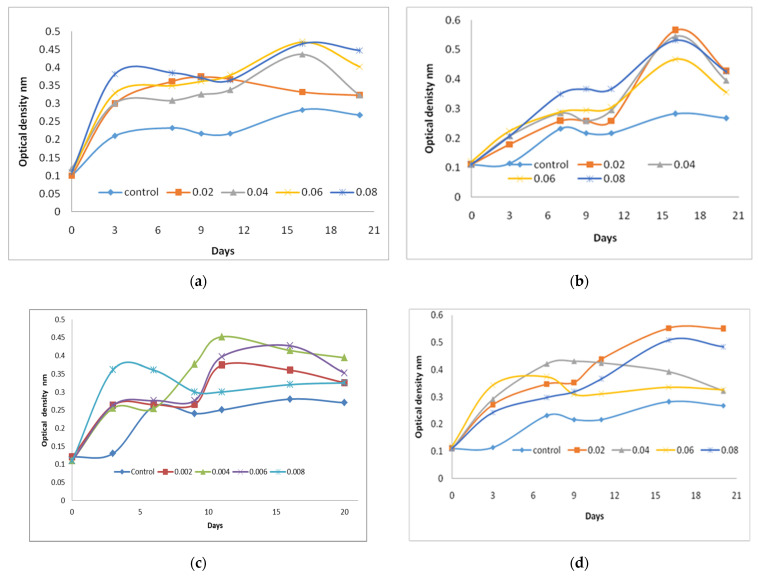
Effect of different concentrations of Pyridoxal (vitamin B6) (**a**), Ascorbic acids (vitamin C) (**b**), Thiamine (vitamin B1) (**c**), and riboflavin (vitamin B2) (**d**) on *C. vulgaris* growth measured by Optical density (nm).

**Figure 2 molecules-27-01602-f002:**
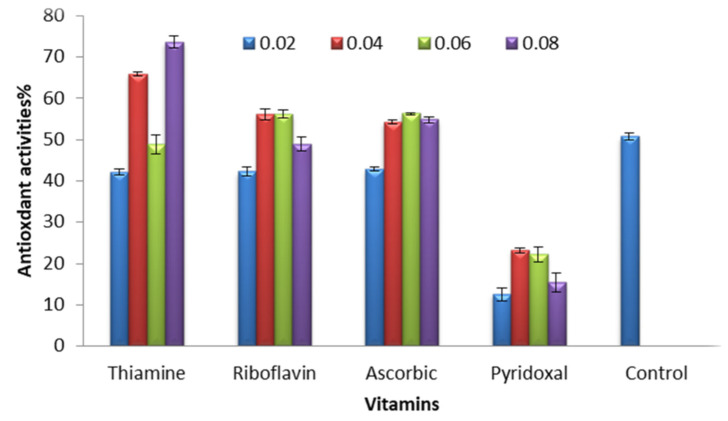
Effects of different concentrations of the tested vitamins supplementation on antioxidant activities of *C. vulgaris* as determined by DPPH (Antioxidant assay) 1,1-Diphenyl-2-picryl-hydrazyl.

**Figure 3 molecules-27-01602-f003:**
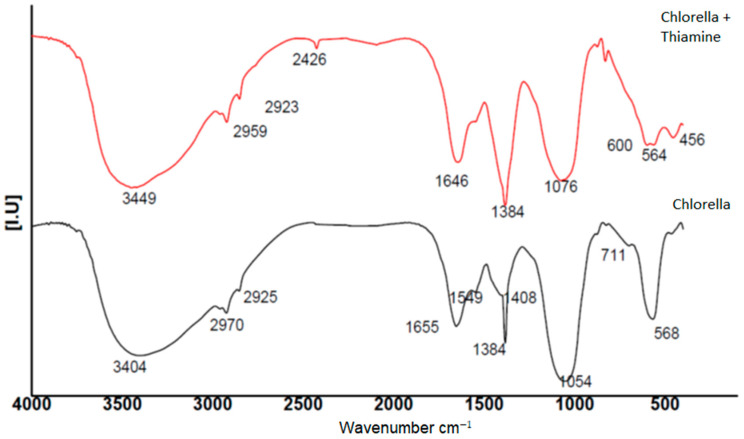
The FT-IR frequency range and the following functional groups are present in the *C. vulgaris* and with thiamine.

**Table 1 molecules-27-01602-t001:** Pigments produced by *C. vulgaris* supplemented with various concentrations of Pyridoxal (vitamin B6).

Vitamins Conc. µg/L	Pigments	Days						
		3	7	9	11	16	20	*p*-Value
Control	Chl a	1.17 ± 0.06 bB	3.12 ± 0.02 aB	1.53 ± 0.08 bB	3.06 ± 0.49 aB	1.81 ± 0.05 bB	1.79 ± 0.07 bB	**
	Chl b	1.04 ± 0.05 B	1.28 ± 0.09 C	1.26 ± 0.06 B	1.12 ± 0.35 B	1.61 ± 0.14 B	1.23 ± 0.04 B	NS
	Carotenoid	10.65 ± 1.50 cA	5.36 ± 0.29 dA	10.88 ± 0.88 cbA	17.48 ± 0.83 aA	16.93 ± 0.29 aA	13.53 ± 0.83 bA	**
*p*-Value		***	***	***	***	***	***	
0.02	Chl a	2.25 ± 0.48 abB	1.63 ± 0.20 bB	1.55 ± 0.03 bB	2.83 ± 0.38 aB	2.03 ± 0.27 abB	2.24 ± 0.09 abB	*
	Chl b	0.93 ± 0.17 bB	1.96 ± 0.26 bB	2.10 ± 0.04 bB	3.63 ± 0.43 bB	6.42 ± 1.69 aB	3.50 ± 0.43 bB	**
	Carotenoid	10.50 ± 1.24 bA	7.75 ± 0.98 bA	12.68 ± 0.34 bA	24.12 ± 2.73 aA	19.21 ± 2.96 abA	12.42 ± 1.85 bA	***
*p*-Value		***	***	***	***	***	***	
0.04	Chl a	0.54 ± 0.14 cB	1.30 ± 0.04 cbB	1.24 ± 0.24 cbB	2.57 ± 0.68 aB	2.85 ± 0.36 abB	2.27 ± 0.59 abB	**
	Chl b	0.92 ± 0.37 cB	1.45 ± 0.28 cbB	1.43 ± 0.009 cbB	2.81 ± 0.47 aB	2.32 ± 0.18 abB	1.38 ± 0.50 cbB	*
	Carotenoid	9.19 ± 1.64 bA	10.24 ± 1.72 bA	10.32 ± 0.95 bA	19.60 ± 1.50 aA	15.42 ± 1.10 aA	17.89 ± 2.32 aA	***
*p*-Value		***	***	***	***	***	***	
0.06	Chl a	0.28 ± 0.03 dC	1.03 ± 0.07 cdB	1.55 ± 0.08 cbB	2.58 ± 0.55 aB	2.18 ± 0.33 abB	1.81 ± 0.14 abcB	**
	Chl b	2.06 ± 0.63 abB	1.46 ± 0.10 bB	2.23 ± 0.17 abB	2.54 ± 0.63 abB	2.92 ± 0.122 aB	2.51 ± 0.35 abB	*
	Carotenoid	6.80 ± 0.47 bA	12.31 ± 0.54 abA	12.03 ± 0.59 abA	12.10 ± 4.18 abA	16.71 ± 1.39 aA	15.43 ± 0.79 aA	***
*p*-Value		***	***	***	***	***	***	
0.08	Chl a	0.19 ± 0.06 cB	2.58 ± 0.43 aB	1.58 ± 0.62 abB	1.41 ± 0.31 bB	1.81 ± 0.60 abB	1.64 ± 0.08 abB	***
	Chl b	0.84 ± 0.40 dB	1.17 ± 0.09 cdB	1.99 ± 0.05 cbB	2.45 ± 0.36 bB	3.63 ± 0.65 aB	2.72 ± 0.065 abB	***
	Carotenoid	5.20 ± 1.55 bA	9.68 ± 1.53 abA	11.25 ± 1.46 abA	16.85 ± 4.50 aA	11.28 ± 2.20 abA	9.32 ± 1.30 abA	*
*p*-Value		***	***	***	***	***	***	

Data are expressed as (mean values ± S.E.), rows showing different small letters (a, b, c) and column within each concentration showing different capital letters (A, B, C) are significantly different at (*p* ≤ 0.05). * *p* < 0.05; ** *p* < 0.01; *** *p* < 0.001. NS: non-significant.

**Table 2 molecules-27-01602-t002:** Pigments produced by *C. vulgaris* supplemented with various concentrations of Ascorbic acid (Vitamin C).

Vitamin Conc. µg/L	Pigments	Days						
		3	7	9	11	16	20	*p*-Value
Control	Chl a	1.17 ± 0.06 bB	3.12 ± 0.02 aB	1.53 ± 0.08 bB	3.06 ± 0.49 aB	1.81 ± 0.05 bB	1.79 ± 0.07 bB	**
	Chl b	1.04 ± 0.05 B	1.28 ± 0.09 C	1.26 ± 0.06 B	1.12 ± 0.35 B	1.61 ± 0.14 B	1.23 ± 0.04 B	NS
	Carotenoid	10.65 ± 1.50 cA	5.36 ± 0.29 dA	10.88 ± 0.88 cbA	17.48 ± 0.83 aA	16.93 ± 0.29 aA	13.53 ± 0.83 bA	**
*p*-Value		***	***	***	***	***	***	
0.02	Chl a	2.08 ± 0.06 cAB	2.82 ± 0.22 cbB	2.39 ± 0.23 cbB	4.02 ± 0.37 abB	5.65 ± 1.06 abB	4.13 ± 0.68 aB	**
	Chl b	0.36 ± 0.189 cB	1.79 ± 0.13 abB	1.58 ± 0.06 abB	2.14 ± 0.71 aC	0.90 ± 0.13 bcC	1.27 ± 0.35 abcC	*
	Carotenoid	9.65 ± 4.15 bA	9.12 ± 2.59 bA	10.12 ± 1.73 bA	16.19 ± 0.08 abA	22.91 ± 1.82 aA	15.35 ± 0.49 bA	**
*p*-Value		*	*	***	***	***	***	
0.04	Chl a	1.13 ± 0.43 bB	2.42 ± 0.25 abB	2.43 ± 0.32 abB	1.71 ± 1.01 bB	4.52 ± 1.48 aAB	2.29 ± 0.28 abB	*
	Chl b	0.67 ± 0.10 bB	1.55 ± 0.61 bB	1.64 ± 0.054 bB	4.56 ± 1.05 aB	2.07 ± 0.05 bB	1.76 ± 0.06 bB	***
	Carotenoid	11.92 ± 2.08 abA	8.86 ± 1.53 bA	11.51 ± 1.06 abA	19.62 ± 2.98 aA	13.87 ± 5.19 abA	10.34 ± 2.55 abA	*
*p*-Value		***	**	***	***	*	**	
0.06	Chl a	1.10 ± 0.54 B	2.87 ± 0.89 B	1.78 ± 0.07 B	1.96 ± 0.44 B	1.93 ± 0.79 B	2.14 ± 0.53 B	NS
	Chl b	0.62 ± 0.31 B	1.84 ± 0.56 B	1.83 ± 0.09 B	2.65 ± 0.81 B	2.52 ± 0.93 B	1.69 ± 0.41 B	NS
	Carotenoid	14.38 ± 6.28 A	8.30 ± 1.71 A	11.30 ± 0.86 A	16.83 ± 0.40 A	16.04 ± 1.004 A	10.75 ± 2.15 A	NS
*p*-Value		*	**	***	***	***	**	
0.08	Chl a	1.26 ± 0.50 B	2.75 ± 0.57 B	2.07 ± 0.07 B	2.45 ± 0.88 B	2.72 ± 0.35 B	2.65 ± 0.33 B	NS
	Chl b	0.36 ± 0.138 dB	1.76 ± 0.36 bcB	1.32 ± 0.12 cB	2.90 ± 0.53 aB	2.66 ± 0.22 abB	1.09 ± 0.14 cdB	***
	Carotenoid	13.50 ± 4.50 A	11.01 ± 0.53 A	11.29 ± 1.13 A	17.97 ± 2.64 A	15.20 ± 0.77 A	14.17 ± 1.33 A	NS
*p*-Value		*	***	***	***	***	***	

Data are expressed as (mean values ± S.E.), rows showing different small letters (a, b, c) and column within each concentration showing different capital letters (A, B, C) are significantly different at (*p* ≤ 0.05). * *p* < 0.05; ** *p* < 0.01; *** *p* < 0.001. NS: non-significant.

**Table 3 molecules-27-01602-t003:** Pigments produced by *C. vulgaris* supplemented with various concentrations of Thiamine (vitamin B1).

Vitamin Conc. µg/L	Pigments	Days						
		3	7	9	11	16	20	*p*-Value
Control	Chl *a*	1.17 ± 0.06 bB	3.12 ± 0.02 aB	1.53 ± 0.08 bB	3.06 ± 0.49 aB	1.81 ± 0.05 bB	1.79 ± 0.07 bB	**
	Chl b	1.04 ± 0.05 B	1.28 ± 0.09 C	1.26 ± 0.06 B	1.12 ± 0.35 B	1.61 ± 0.14 B	1.23 ± 0.04 B	NS
	Carotenoid	10.65 ± 1.50 cA	5.36 ± 0.29 dA	10.88 ± 0.88 cbA	17.48 ± 0.83 aA	16.93 ± 0.29 aA	13.53 ± 0.83 bA	**
*p*-Value		***	***	***	***	***	***	
0.02	Chl a	0.63 ± 0.21 d	2.26 ± 0.34 cbB	1.28 ± 0.19 cdB	4.28 ± 0.56 aB	3.41 ± 0.90 ab	3.26 ± 0.32 abB	**
	Chl b	0.08 ± 0.13 d	2.17 ± 0.42 cbB	1.51 ± 0.07 cB	4.26 ± 0.78 aB	3.47 ± 0.58 ab	2.20 ± 0.18 bcB	*
	Carotenoid	9.05 ± 5.32	8.22 ± 1.51 A	14.68 ± 0.74 A	19.001 ± 3.58 A	9.48 ± 4.53	12.36 ± 0.66 A	NS
*p*-Value		NS	***	***	***	NS	***	
0.04	Chl a	0.42 ± 0.37 b	2.29 ± 0.03 abB	2.02 ± 0.035 abB	3.64 ± 1.58 aB	4.01 ± 0.43 a	4.05 ± 0.63 aB	**
	Chl b	1.24 ± 0.89	2.50 ± 0.31 B	1.54 ± 0.09 B	2.85 ± 0.61 B	3.07 ± 0.60	1.54 ± 0.27 C	NS
	Carotenoid	3.77 ± 2.18 b	7.30 ± 0.59 abA	9.80 ± 1.78 abA	15.60 ± 2.49 aA	8.94 ± 4.84 ab	7.56 ± 0.78 abA	**
*p*-Value		NS	**	**	***	NS	**	
0.06	Chl a	0.76 ± 0.27 bB	2.42 ± 0.34 aB	2.02 ± 0.039 abB	2.68 ± 0.57 a	2.05 ± 0.78 ab	2.15 ± 0.11 abB	**
	Chl b	0.93 ± 0.62 cB	1.93 ± 0.23 cbB	2.08 ± 0.14 cbB	3.85 ± 0.41 a	3.09 ± 0.60 ab	2.23 ± 0.29 bcB	**
	Carotenoid	10.54 ± 2.00 A	10.96 ± 1.77 A	8.57 ± 1.23 A	7.48 ± 4.09	4.66 ± 0.92	8.86 ± 1.44 A	NS
*p*-Value		***	***	***	NS	NS	***	
0.08	Chl a	0.036 ± 0.15 c	3.47 ± 0.172 aB	1.98 ± 0.109 bB	1.78 ± 0.400 bB	1.93 ± 0.50 bB	1.66 ± 0.46 bB	***
	Chl b	2.23 ± 0.96	1.74 ± 0.08 B	2.10 ± 0.20 B	2.92 ± 0.18 B	2.24 ± 0.39 B	2.14 ± 0.52 B	NS
	Carotenoid	5.34 ± 2.83 b	8.97 ± 1.001 bA	10.83 ± 3.05 bA	29.27 ± 2.40 aA	12.43 ± 3.02 bA	8.11 ± 0.47 bA	***
*p*-Value		*	**	**	***	***	***	

Data are expressed as (mean values ± S.E.), rows showing different small letters (a, b, c) and column within each concentration showing different capital letters (A, B, C) are significantly different at (*p* ≤ 0.05). * *p* < 0.05; ** *p* < 0.01; *** *p* < 0.001. NS: non-significant.

**Table 4 molecules-27-01602-t004:** Pigments produced by *C. vulgaris* supplemented with various concentrations of riboflavin (vitamin B2).

Vitamin Conc. µg/L	Pigments	Days						
		3	7	9	11	16	20	*p*-Value
Control	Chl a	1.17 ± 0.06 bB	3.12 ± 0.02 aB	1.53 ± 0.08 bB	3.06 ± 0.49 aB	1.81 ± 0.05 bB	1.79 ± 0.07 bB	**
	Chl b	1.04 ± 0.05 B	1.28 ± 0.09 C	1.26 ± 0.06 B	1.12 ± 0.35 B	1.61 ± 0.14 B	1.23 ± 0.04 B	NS
	Carotenoid	10.65 ± 1.50 cA	5.36 ± 0.29 dA	10.88 ± 0.88 cbA	17.48 ± 0.83 aA	16.93 ± 0.29 aA	13.53 ± 0.83 bA	**
*p*-Value		***	***	***	***	***	***	
0.02	Chl a	0.688 ± 0.05 cB	2.82 ± 0.52 b	1.92 ± 0.37 cb	5.30 ± 0.55 aA	2.92 ± 0.41 b	2.92 ± 0.558 bA	***
	Chl b	1.91 ± 0.06 aA	1.69 ± 0.14 ab	1.36 ± 0.226 ab	1.58 ± 0.44 abB	1.77 ± 0.36 ab	0.93 ± 0.197 bB	*
	Carotenoid	0.68 ± 0.05 cB	2.82 ± 0.526 b	1.92 ± 0.37 cb	5.31 ± 0.55 aA	2.92 ± 0.41 b	2.92 ± 0.55 bA	***
*p*-Value		**	NS	NS	**	NS	**	
0.04	Chl a	0.353 ± 0.03 c	2.88 ± 0.75 a	2.17 ± 0.44 ab	1.41 ± 0.35 bc	2.20 ± 0.20 ab	1.98 ± 0.17 ab	***
	Chl b	0.40 ± 0.07 c	1.49 ± 0.56 cb	1.98 ± 0.38 ab	3.11 ± 0.84 a	2.37 ± 0.15 ab	1.56 ± 0.25 bc	***
	Carotenoid	0.35 ± 0.03 c	2.88 ± 0.75 a	2.17 ± 0.44 ab	1.41 ± 0.35 bc	2.20 ± 0.20 ab	1.98 ± 0.17 ab	***
*p*-Value		NS	NS	NS	NS	NS	NS	
0.06	Chl a	0.04 ± 0.12 bB	4.93 ± 2.87 a	1.67 ± 0.169 ab	2.60 ± 0.85 ab	1.81 ± 0.33 abB	1.70 ± 0.16 ab	***
	Chl b	1.51 ± 0.52 cbA	1.13 ± 0.94 c	1.81 ± 0.197 cb	4.48 ± 1.08 a	3.50 ± 0.11 abA	2.19 ± 0.36 bc	**
	Carotenoid	0.04 ± 0.12 bB	4.93 ± 2.87 a	1.67 ± 0.16 ab	2.61 ± 0.85 ab	1.81 ± 0.33 abB	1.70 ± 0.16 ab	***
*p*-Value		**	NS	NS	NS	***	NS	
0.08	Chl a	0.44 ± 0.27 c	1.81 ± 0.35 b	1.47 ± 0.132 bB	2.99 ± 0.47 a	2.15 ± 0.25 abB	1.88 ± 0.12 b	**
	Chl b	1.04 ± 0.38 c	2.26 ± 0.13 b	2.63 ± 0.01 abA	3.49 ± 0.50 a	3.15 ± 0.05 abA	2.16 ± 0.49 b	**
	Carotenoid	0.44 ± 0.27 c	1.81 ± 0.35 b	1.47 ± 0.13 bB	2.99 ± 0.47 a	2.15 ± 0.25 abB	1.88 ± 0.12 b	**
*p*-Value		NS	NS	**	NS	*	NS	***

Data are expressed as (mean values ± S.E.), rows showing different small letters (a, b, c) and column within each concentration showing different capital letters (A, B, C) are significantly different at (*p* ≤ 0.05). * *p* < 0.05; ** *p* < 0.01; *** *p* < 0.001. NS: non-significant.

**Table 5 molecules-27-01602-t005:** Effect of different concentrations of the tested vitamins on carbohydrate and proteins contents of *C. vulgaris*.

Vitamin	Concentration (µg/L)	Carbohydrate (mg/g)	Proteins (mg/g)
Control	0	149.97 ± 0.88	50.676 ± 0.864
Thiamine	0.02	165.30 ± 0.57 c	55.088 ± 1.452 b
	0.04	147.64 ± 1.76 d	48.552 ± 0.589 c
	0.06	195.30 ± 0.57 b	52.637 ± 2.536 cb
	0.08	250.30 ± 0.57 a	64.402 ± 0.712 a
	*p*-Value	***	***
Riboflavin	0.02	194.64 ± 0.33 c	49.533 ± 0.993 b
	0.04	171.30 ± 0.57 d	46.755 ± 1.337 b
	0.06	207.97 ± 1.45 b	63.912 ± 0.993 a
	0.08	215.97 ± 1.20 a	47.572 ± 1.609 b
	*p*-Value	***	***
Asorbic acid	0.02	206.30 ± 1.15 c	63.748 ± 0.432 a
	0.04	244.64 ± 0.33 b	48.552 ± 0.589 d
	0.06	206.64 ± 0.88 c	50.350 ± 0.283 c
	0.08	284.64 ± 0.33 a	53.781 ± 0.748 b
	*p*-Value	***	***
Pyridoxal	0.02	244.64 ± 0.33 c	45.774 ± 1.452 b
	0.04	224.64 ± 0.33 d	43.650 ± 0.589 b
	0.06	345.30 ± 0.57 b	56.069 ± 1.884 a
	0.08	348.97 ± 1.85 a	45.284 ± 2.305 b
	*p*-Value	***	***

Data are expressed as (mean values ± S.E.), column showing different small letters (a, b, c) are significantly different at (*p* ≤ 0.05). *** *p* < 0.001.

**Table 6 molecules-27-01602-t006:** Percentage of relative inhibitions of various cancer cells (HEPG-2, HCT-116, Hela, and PC-3) that treated by different concentrations of methanol extracts of C. vulgaris (Chl) and *C. vulgaris* supplemented with Thiamine (vitamin B1) (Chl + Thia). Human prostate cancer (PC-3), Hepatocellular carcinoma (HEPG-2), Colorectal carcinoma (HCT-116), and Epitheliod Carcinoma (Hela).

Cancer Cell	HePG-2		HCT-116		Hela		PC3	
Conc., µg/mL	Chl	Chl + Thia	Chl	Chl + Thia	Chl	Chl + Thia	Chl	Chl + Thia
100	71.2 ± 4	83.8 ± 2	68.9 ± 6	89.3 ± 5	80.1 ± 9	88.4 ± 4	64 ± 4	84.8 ± 6
200	83.8 ± 5	89.5 ± 5	76.5 ± 8	93.8 ± 4	90.2±	90.7 ± 9	80.3 ± 5	91.3 ± 4
400	90.9 ± 9	92.6 ± 6	87.7 ± 4	97.7 ± 7	95.5 ± 3	95 ± 7	88.8 ± 4	95.9 ± 3
600	93.6 ± 3	96.4 ± 4	95.1 ± 3	98.2 ± 8	97.4 ± 8	97.8 ± 4	92.7 ± 3	96.9 ± 7
800	95.7 ± 6	97.9 ± 9	97.6 ± 5	99 ± 4	98.1 ± 4	98.8 ± 8	94.4 ± 3	98.2 ± 4

**Table 7 molecules-27-01602-t007:** BG11 medium contents.

Constituent A		Constituent C	
Stock solution	g/L	Trace elements	mg/L
NaNO_3_	1.5	H_3_BO_3_	2.8
K_2_HPO_4_·3H_2_O	0.04	MnCl_2_·4H_2_O	1.81
MgSO_4_·7H_2_O	0.075	ZnSO_4_·7H_2_O	0.222
CaCl_2_·2H_2_O	0.036	Na·MoO_4_·2H_2_O	0.39
Na_2_CO_3_	0.02	CuSO_4_·H_2_O	0.079
**Constituent B**		Co(NO_3_)_2_·6H_2_O	0.0494
EDTA (disodium magnesium salt)	0.001 g/L		

## Data Availability

Not applicable.

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
