# Peer review of "Assessment of Antioxidant and Anticancer Activities of Microgreen Alga Chlorella vulgaris and Its Blend with Different Vitamins"

_molecules, 2022, doi:10.3390/molecules27051602_

Round 1

Reviewer 1 Report

Dear authors,

Please see the comments below.

The article is suitable for Molecules. The experimental part of the study was well designed and had proper analytical tools and data analysis.

Editing suggestions: Lines: 95, 105, 121, 156, 162, 163, 170, 176, 188, 193, 202 add spacing and/or properly punctuations.

The name C. vulgaris was not always written in italics (lines: 99, 113, 115, 117, 136, 154, 188, 191, 215, 217, 218, 221, 320).

There were inconsistencies in writing unit for volume or concentrations (lines:195, 334-335, 353).

Corrections in the text:  name for tested vitamins in Figure 2 (incorrect: thamine, asorbic), line: 369 name for DPPH activity.

Also, looking at Figure 1(c) is not visible the legend for all concentrations.

I suggest improving the discussions about antioxidant and anticancer effects. I think the goal of the article could be highlighted by this.

Author Response

We would like to send a sincere thanks to the editor for handling the manuscript and for the comments and suggestions and for taking the time and energy to help us improve the paper.  We agreed with all comments and suggestions. The manuscript has been revised following suggestions of the Reviewers' comments carefully. We hope that these revisions improve the manuscript, now find it worthy of publication.

Review 1

The article is suitable for Molecules. The experimental part of the study was well designed and had proper analytical tools and data analysis.

Many thanks

Editing suggestions: Lines: 95, 105, 121, 156, 162, 163, 170, 176, 188, 193, 202 add spacing and/or properly punctuations.

Response

The spacing added or deleted extra spacing

The name C. vulgaris was not always written in italics (lines: 99, 113, 115, 117, 136, 154, 188, 191, 215, 217, 218, 221, 320).

Response

Changed

There were inconsistencies in writing unit for volume or concentrations (lines:195, 334-335, 353).

Response

Corrected

Corrections in the text:  name for tested vitamins in Figure 2 (incorrect: thamine, asorbic), line: 369 name for DPPH activity.

Response

Corrected

Also, looking at Figure 1(c) is not visible the legend for all concentrations.

Response

Corrected

I suggest improving the discussions about antioxidant and anticancer effects. I think the goal of the article could be highlighted by this.

Response

The following reference added

  1. Malanga, G., & Puntarulo, S. (1995). Oxidative stress and antioxidant content in Chlorella vulgaris after exposure to ultraviolet‐B radiation. Physiologia plantarum, 94(4), 672-679.‏
  2. Miranda MS, Sato S, Mancini-Filho J. Antioxidant activity of the microalga Chlorella vulgaris cultered on special conditions. Bollettino Chimico Farmaceutico. 2001 May-Jun;140(3):165-168. PMID: 11486607.
  3. Wang, H. Y., Zeng, X. B., Guo, S. Y., & Li, Z. T. (2008). Effects of magnetic field on the antioxidant defense system of recirculation‐cultured Chlorella vulgaris. Bioelectromagnetics: Journal of the Bioelectromagnetics Society, The Society for Physical Regulation in Biology and Medicine, The European Bioelectromagnetics Association, 29(1), 39-46.‏
  4. El-fayoumy, E.A., Shanab, S.M.M., Gaballa, H.S. et al. Evaluation of antioxidant and anticancer activity of crude extract and different fractions of Chlorella vulgaris axenic culture grown under various concentrations of copper ions. BMC Complement Med Ther 21, 51 (2021). https://doi.org/10.1186/s12906-020-03194-x
  5. Yusof, Y. A. M., Saad, S. M., Makpol, S., Shamaan, N. A., & Ngah, W. Z. W. (2010). Hot water extract of Chlorella vulgaris induced DNA damage and apoptosis. Clinics, 65(12), 1371-1377.‏
  6. Hamouda, R. A., Abd El Maksoud, A. I., Wageed, M., Alotaibi, A. S., Elebeedy, D., Khalil, H., ... & Abdella, A. (2021). Characterization and Anticancer Activity of Biosynthesized Au/Cellulose Nanocomposite from Chlorella vulgaris. Polymers, 13(19), 3340.‏

Reviewer 2 Report

The submitted manuscript's results are really interesting, yet, heavy English editing and grammar checking is needed. This includes the style, the name of the chemicals etc.

Also, the figures' resolutions must be improved and also their visibility is low, please use multiple, significantly different colours.

Tables should be revised, reduced in size or separated into more tables.

In generals, the Methods part is missing many detailed content, please revise it. All descriptions must be as long and detailed to let the reader to repeat the experiments.

What was the effect of methanol on the cells? How were the cell viability values calculated? Compared to the untreated control or compared to the methanol treated cells?

What are the components of BG 11 media? Why isn't it a standard procedure to supplement it with vitamins?

Author Response

The submitted manuscript's results are really interesting, yet, heavy English editing and grammar checking is needed. This includes the style, the name of the chemicals etc.

Response

The English editing improved and the chemical name corrected

Also, the figures' resolutions must be improved and also their visibility is low, please use multiple, significantly different colours.

Response

Figure 1 has different colour and figure 2 improved

Tables should be revised, reduced in size or separated into more tables.

Response

I agree with you but I can not to separate table in many tables because the major data present in the table.

In generals, the Methods part is missing many detailed content, please revise it. All descriptions must be as long and detailed to let the reader to repeat the experiments.

Response

We agree, added more detailed and  references

What was the effect of methanol on the cells? How were the cell viability values calculated? Compared to the untreated control or compared to the methanol treated cells?

Response

Added to methods the following,

The cell lines declared above were used to investigate the suppression effects of methanol extracts of C. vulgaris and C. vulgaris blended with 0.08 µg/L of thiamine on cell growth using the MTT assay [58, 59]. (the methanol extracts of C. vulgaris and C. vulgaris blended with 0.08 µg/L of thiamine were evaporated by Rotary evaporator, followed by dissolved in Dimethyl sulfoxide (DMSO) to prepare the following concentration (100, 200, 400, 600, and 800 µg/ml)). The colorimetric assay is measured at absorbance of 570 nm using a plate reader (EXL 800 , USA). The relative cell viability in percentage was calculated as (A570 of treated samples/A570 of untreated sample) X 100

What are the components of BG 11 media? Why isn't it a standard procedure to supplement it with vitamins?

Response

The BG11 medium added to the manuscript,

The control is present (Chlorella vulgaris without addition of vitamins)

Reviewer 3 Report

According to my opinion, the manuscript entitled "Assessment of antioxidant and anticancer activities of micro-2 green alga Chlorella vulgaris and its blend with different vitamins" given by Ragaa A Hamouda, Amera Abd El Latif, Ebtihal M.Elkaw, Amenah S. Al-otaibi and Hanafy A. Hamza is appropriate for Molecules. I predict that the results of the examinations presented in this manuscript will be attractive for a wide audience of scientist, i.e. chemists and pharmacists and biologists. The results will be especially attractive for scientists  whose scientific interests focus around antioxidant and anticancer activity of various natural materials and their blend with vitamins. The experimental part of this manuscript is adequate to the idea of this research. In summary, I recommend to accept this manuscript in the current form.

Author Response

Review 3

According to my opinion, the manuscript entitled "Assessment of antioxidant and anticancer activities of micro-2 green alga Chlorella vulgaris and its blend with different vitamins" given by Ragaa A Hamouda, Amera Abd El Latif, Ebtihal M.Elkaw, Amenah S. Al-otaibi and Hanafy A. Hamza is appropriate for Molecules. I predict that the results of the examinations presented in this manuscript will be attractive for a wide audience of scientist, i.e. chemists and pharmacists and biologists. The results will be especially attractive for scientists  whose scientific interests focus around antioxidant and anticancer activity of various natural materials and their blend with vitamins. The experimental part of this manuscript is adequate to the idea of this research. In summary, I recommend to accept this manuscript in the current form.

Response

Many thanks

Reviewer 4 Report

The manuscript by Hamouda et al. reports an effect of different vitamins added in the culture of Chlorella vulgaris on biological activity of its extract as well as gives some incites in the chemical composition of the extract. Though the research design and the quality of the data is acceptable, the manuscript at the end accesses the biological effect of the complex mixture (i.e. extract) and lacks evaluation of the effect of individual compounds and characterization of these active compounds. Thus, the manuscript lacks evaluation of the chemical  composition and / or isolation of individual compounds and it is therefore not clear what substance is responsible for the observed biological effect. Therefore, I feel that this work is more suitable for another journal from the same section (Microalgal Biotechnology for Bioproducts and Food Applications), which has more biotechnological (or ecological), rather them chemical scope. For example “Foods” or “Phycology”.

Additionally, the Methods must be described in more detail.

Author Response

The manuscript by Hamouda et al. reports an effect of different vitamins added in the culture of Chlorella vulgaris on biological activity of its extract as well as gives some incites in the chemical composition of the extract. Though the research design and the quality of the data is acceptable, the manuscript at the end accesses the biological effect of the complex mixture (i.e. extract) and lacks evaluation of the effect of individual compounds and characterization of these active compounds. Thus, the manuscript lacks evaluation of the chemical  composition and / or isolation of individual compounds and it is therefore not clear what substance is responsible for the observed biological effect. Therefore, I feel that this work is more suitable for another journal from the same section (Microalgal Biotechnology for Bioproducts and Food Applications), which has more biotechnological (or ecological), rather them chemical scope. For example “Foods” or “Phycology”.

Response

The manuscript proved the effect of crude extract of alga and crude extract of alga supplemented with vitamins, on the pigments, and antioxidant activities, protein and carbohydrate contents, so we have a large data. Next research we will focus in the separation of compounds that present in crude oil.

Additionally, the Methods must be described in more detail.

Response

 The methods now cleared and connected with references to obtain more description

Round 2

Reviewer 2 Report

Dear Authors,

The quality of the resubmitted manuscript is highly improved and most of the issues were solved. Thank you for your hard work.

My only remaining problem is the formatting of the subfigures of Figure 1. All Figures should have the same font size, resolution. For example, in case of Figure b and c, the legends touch the actual graph. Please correct it.